# MULTI-ATTACKS: A SINGLE ADVERSARIAL PERTURBATION FOR MULTIPLE IMAGES AND TARGET LABELS

## ABSTRACT

We show that we can easily design a single adversarial perturbation $P$ that changes the class of $n$ images $X_1, X_2, \ldots, X_n$ from their original, unperturbed classes $c_1, c_2, \ldots, c_n$ to desired (not necessarily all the same) classes $c_1^*, c_2^*, \ldots, c_n^*$ for up to hundreds of images and target classes at once. We call these *multi-attacks*. Characterizing the maximum $n$ we can achieve under different conditions such as image resolution, we estimate the number of regions of high class confidence around a particular image in the space of pixels to be around $10^{\mathcal{O}(100)}$, posing a significant problem for exhaustive defense strategies. We show several immediate consequences of this: adversarial attacks that change the resulting class based on their intensity, and scale-independent adversarial examples. To demonstrate the redundancy and richness of class decision in the pixel space, we look for its two-dimensional sections that trace images and spell words using particular classes. We also show that ensembling reduces susceptibility to multi-attacks, and that classifiers trained on random labels are more susceptible.

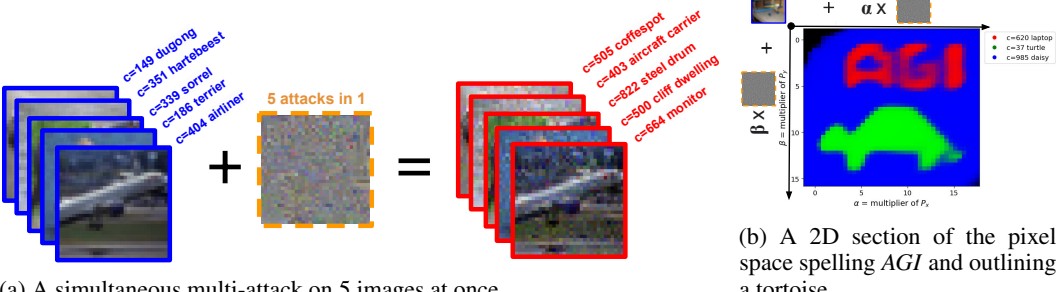

(a) A simultaneous multi-attack on 5 images at once.

(b) A 2D section of the pixel space spelling *AGI* and outlining a tortoise.

Figure 1: (**Left panel**): A single, small adversarial perturbation (in orange) to the pixels of many images at once (5 in this example, in blue) can change their classification to arbitrarily chosen classes (that are not necessarily the same). The partitioning of the space of inputs into classes is extremely rich and redundant, and allows for optimization of very constrained problems, such as finding a simultaneous attack on $\mathcal{O}(100)$ images at once. (**Right panel**): To demonstrate the richness of the space of inputs, we simultaneous attack 288 neighboring images on a 2D slice of the pixel space (starting at a CIFAR-10 image of a biplane) with 2 adversarial perturbations, $P_x$ and $P_y$, defining a 2D affine subspace around it. We optimize the subspace to spell *AGI* in the ImageNet class 620 (laptop) and draw a tortoise in the class 37 (turtle), with a background of class 985 (daisy).

## 1 INTRODUCTION

The problem of adversarial examples is typically framed in the classification setting (Szegedy et al., 2013), where a small, specifically tailored adversarial perturbation $P$ to the input data $X$ changes the classification decision from a class $c$, $c = \operatorname{argmax} f(X)$, to a different class $c^*$, $c^* = \operatorname{argmax} f(X + P)$, while the magnitude of $P$ (typically measured by its $L_2$ or $L_\infty$ norms) remains very small compared to $X$. In other words, the classification decision about an input image $X$ is changed from one class to a confident prediction of another class (chosen by the attacker) by a small, often human-imperceptible change $P$ to the image $X$. This problem is omnipresent in image

classification, starting from small models and data (Szegedy et al., 2013), all the way to current large models such as CLIP (Radford et al., 2021), as described for example in (Fort, 2021). Adversarial vulnerability applies not only to the classification setting but is a prominent issue in other domains, e.g. for near out-of-distribution detection (Chen et al., 2021; Fort, 2022). Despite the tremendous success of deep learning, the persistent presence of adversarial vulnerabilities points towards a potentially deeply ingrained problem with the approach.

In this paper, we investigate the limits of adversarial attacks. The key question we are addressing is whether multiple attacks can be carried out by the same adversarial perturbation. For $n$ images, e.g. an image $X_1$ of a *cat*, an image $X_2$ of a *tortoise*, and an image $X_3$ of a *house*, can we can design a single change $P$ to all $n$ images at once such that $X_1 + P$ is classified as e.g. a *tree*, $X_2 + P$ is a *plane* and $X_3 + P$ is a *dinosaur*?

Our key contribution is to show that such attacks do exist and are in fact easy to find. We call these *multi-attacks*. Multi-attacks are easy to find using standard methods, and the higher the resolution of the image, the more images we can attack at the same time. We demonstrate that models trained on randomly permuted labels are more susceptible to multi-attacks, and that ensembling multiple models decreases their susceptibility to multi-attacks. Using a simple toy model theory, we estimate the number of distinct class regions around each image in the space of pixels to be $10^{\mathcal{O}(100)}$, which poses a major challenge to any adversarial defence strategies that rely on exhaustion. To show the flexibility and richness provided by the $10^{\mathcal{O}(100)}$ class regions around each image in the space of pixels, we show that we can easily find two-dimensional sections of the input space that show images and spell words in arbitrary classes.

## 2 METHOD

For a classifier $f : X \to y$, $n$ inputs $X_1, X_2, \ldots, X_n$, and $n$ target classes $c_1^*, c_2^*, \ldots, c_n^*$, the goal is to produce an adversarial perturbation $P$ such that $\operatorname{argmax} f(X_1 + P) = c_1^*$, $\operatorname{argmax} f(X_2 + P) = c_2^*$, $\ldots \operatorname{argmax} f(X_n + P) = c_n^*$. A standard adversarial attack takes a single image $X$ and finds a single perturbation $P$ such that $X + P$ is misclassified as the target class the attacker chose. In this paper, we are looking for *single* perturbation $P$ that is simultaneously capable of changing *many* images to *many* distinct classes. We call these perturbations *multi-attacks*.

### 2.1 GENERATING A MULTI-ATTACK

To find a multi-attack $P$, we are using the simplest method available. Given $n$ target labels $y_i^* \in \{0, 1, \ldots, C - 1\}^n$, and $n$ input images $X \in \{X_1, X_2, \ldots, X_n\}$, we get the classifier logits $z_i = f(X_i)$ for each image, and compute the cross-entropy loss against the desired target labels as

$$\mathcal{L} = \frac{1}{n} \sum_{i=0}^{n-1} \mathrm{CE}(z_i, y_i^*), \tag{1}$$

where CE is the standard cross-entropy loss

$$\mathrm{CE}(z, y) = - \sum_{j=0}^{C-1} y_j \log \left( \frac{\exp z_j}{\sum_{k=0}^{C-1} \exp z_k} \right), \tag{2}$$

and $C$ is the total number of classes. We use the Adam optimizer (Kingma & Ba, 2014) to get the attack $P$ by using the gradient of the loss $\mathcal{L}$ with respect to the perturbation $P$. This is the same way adversarial examples, first described in (Szegedy et al., 2013), were generated. More robust methods exist, such as *Fast Gradient Sign Method* in (Goodfellow et al., 2014) that only uses the signs of the gradient instead of the gradient itself, however, we found that simply taking the gradient itself and doing the most straightforward thing worked well enough.

Starting from the input images $X$ of the shape $[\mathrm{batch}, \mathrm{channels}, \mathrm{resolution}, \mathrm{resolution}]$ and the target labels $y^*$ of the shape $[\mathrm{batch}]$, we are gradually getting updates to the perturbation $P$ of the shape $[1, \mathrm{channels}, \mathrm{resolution}, \mathrm{resolution}]$. The standard gradient descent approach would look like

$$P_{t+1} = P_t - \eta \frac{\partial \mathcal{L}(f(X + P), y^*)}{\partial P} \Big|_{P = P_t}, \tag{3}$$

where $\eta$ is the learning rate. We are using the Adam optimizer with an arbitrarily chosen learning rate of $10^{-2}$.

## 2.2 ENSEMBLES

Adversarial robustness of classifiers has been shown to improve with the use of ensembles (Tramèr et al., 2020; Kariyappa & Qureshi, 2019). This has also been the case with adversarial attacks against strong out-of-distribution detectors (Fort, 2022). We wanted to see if the number of simultaneously attackable images in a *multi-attack* showed any signs of dependence on the size of an ensemble. To do that, we took models $f_1, f_2, \ldots, f_m$, and averaged their logit outputs to form a single model $f_{\text{ensemble}}(x) = (1/m) \sum_{i=1}^{m} f_i(x)$. The results of these experiments are shown in Section 4.3.

## 3 SIMPLE THEORY

Let us sketch a simple geometric model of the neighborhood of a particular image $X$ in the space of inputs (the space of pixels). For CIFAR-10 (Krizhevsky & Hinton, 2009), this would be a $d_{\text{in}} = 32 \times 32 \times 3 = 3072$ dimensional space of pixels and their channels. Let's imagine that a particular image $X$ is surrounded, within a certain distance, by $N$ cells – regions of a high probability value of some class. We would like to estimate this number $N$. The particular locations of these cells around two inputs $X_1$ and $X_2$ will generically be very different. If we perturb the input $X_1$ by a perturbation $v$, $X_1 + v$ might be a high confidence class 542, while $X_2 + v$ might be in a high confidence region of a wholly different class, or not a high confidence region of any class altogether.

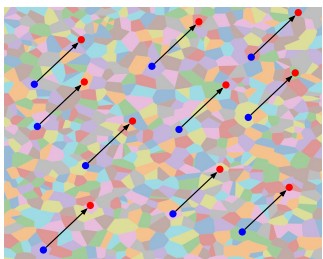

Figure 2: Illustration of the input space partitioning into classes (different colors) and the simultaneous attack (black arrow) on images (blue circles) changing all their classes at once to where their shifted versions (red circles) lie.

For simplicity, let's consider a random perturbation $v$ that for all $X_i + v$ reaches a high-confidence class area for some class. We illustrate the situation in Figure 2. The probability of it reaching the correct class $c_i^*$ on the $i^{\text{th}}$ image is $1/C$, where $C$ is the total number of classes (in our experiments $C = 1000$ since we are using models pretrained on ImageNet (Deng et al., 2009), or 10 for our CIFAR-10 trained models and experiments). To reach the target class for each $i = 1, 2, \ldots, n$, the probability decreases to $(1/C)^n$.

In our toy model, for a vector $v$ to exist such that the predicted classes are $c_1^*, c_2^*, \ldots, c_n^*$, we need there to be at least $N$ regions around each input $X$ where $N(1/C)^n \geq 1$. The maximum number of images we can attack at once, $n_{\text{max}}$, is then approximately

$$n_{\text{max}} \approx \log (N) / \log (C) . \tag{4}$$

Given the maximum number of images we can attack at once, $n_{\text{max}}$, the number of regions surrounding an image is approximately $N \approx \exp (n_{\text{max}} \log(C))$. For $n_{\text{max}} = \mathcal{O}(100)$ and $C = 1000$, this works out to be $N = 10^{\mathcal{O}(100)}$. For CIFAR-10 models with $C = 10$, the difference in the order of magnitude is small.

## 4 EXPERIMENTS

In our experiments, we primarily used an ImageNet (Deng et al., 2009) pretrained ResNet50 (He et al., 2016) from the PyTorch (Paszke et al., 2019) `torchvision` hub[1]. This model has 1000 output classes and uses the $224 \times 224 \times 3$ input resolution. As inputs, we were using images from the CIFAR-10 dataset (Krizhevsky & Hinton, 2009), however, working with random Gaussian noise yielded equivalent results. To study the effect of resolution, we would first change the resolution of the input image to the target resolution $r \times r$, add the attack of the same resolution, and then rescale the result to $224 \times 224$ before feeding it into the classifier. The default learning rate we used

---

[1]`https://pytorch.org/vision/main/models/generated/torchvision.models.resnet50.html`

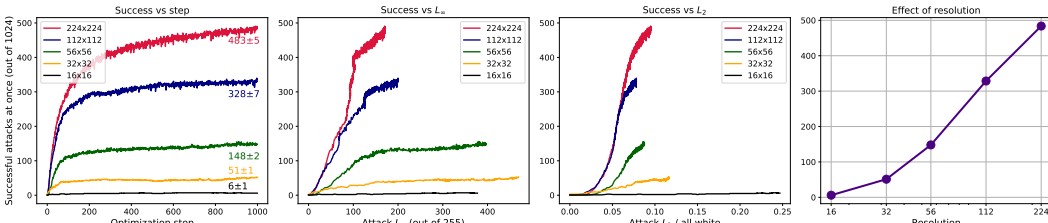

Figure 4: The number of successfully attacked images out of 1024 randomly chosen but fixed samples of CIFAR-10 with 1024 randomly chosen but fixed ImageNet classes (1000 in total) as a function of the size of the adversarial perturbation and optimization step (left-most panel). The higher the resolution of the image, the easier it is to attack a large number of them simultaneously even with a smaller $L_\infty$ norm perturbation.

for finding the adversarial attacks was $10^{-2}$ with the Adam (Kingma & Ba, 2014) optimizer. All experiments were done on a single A100 GPU in a Google Colab in a matter of hours.

To see what the effect of architecture is, we also used a `ResNet18` from the same source. For experiments where we trained models on CIFAR-10 ourselves, we removed the final linear layer from the original architecture and replaced it with a randomly initialized linear layer with 10 outputs. We also trained a small CNN architecture we call a `SimpleCNN` with 4 layers of $3 \times 3$ convolutional kernels followed by `ReLU` and mean pooling, with channel numbers 32, 64, 128, and 128, followed by a linear layer to 10 logits.

## 4.1 NUMBER OF SIMULTANEOUS ATTACKS VS PERTURBATION STRENGTH

We are measuring the strength of the adversarial perturbation by the $L_\infty$ and $L_2$ norms. $L_\infty$ is the maximum value of the perturbation $P$ over all pixels and all channels. We are using pixel values in the 0 to 255 range. Starting with a fixed batch of 1024 images (an arbitrary choice large enough to be challenging at the $224 \times 224$ resolution but also fast enough to experiment with), we track how many have been classified as the randomly chosen but fixed target classes (from the 1000 ImageNet classes) as a function of the iteration of the adversary finding step, and the $L_2$ and $L_\infty$ norms of the attack perturbation. The results are shown in Figure 4. The higher the resolution of the image, the more images we can attack successfully at once. In addition, the $L_\infty$ norms of the higher resolution images are smaller (though still pretty large compared to the standard $8/255$). The $L_2$ norms are roughly equivalent at the end of optimization. Interestingly, it seems (by visual inspection alone) the number of successfully attacked images scales linearly with the logarithm of the resolution, as $n_{\max} \propto \log r$ (as shown in the right-most panel of Figure 4. It is possible that the numbers we obtained is an overestimate of the actual number of images attackable at once to 100% accuracy, since the images that are in some sense easier to attack might be chosen first from the full batch of 1024 images available. We discuss this further in Section 4.2.

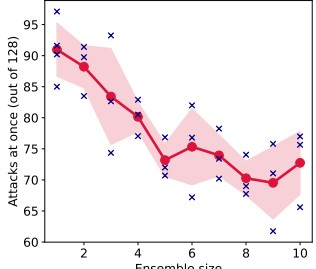

Figure 3: Successfully attacked images out of 128 with a single multi-attack for ensembles of CIFAR-10-trained SimpleCNN models. The larger the ensemble, the fewer images we can successfully attack at the end of the 500 steps of attack optimization.

## 4.2 ATTACKING A BATCH AT ONCE

In our experiments, e.g. in Figure 4, we start with a batch of images (1024 in that Figure), and optimize the cross-entropy of the classifier predictions against the target labels, as shown in Eq. 1. There are two disadvantages to this approach: 1) having the $\arg\max f(X_i + P) = c_i^*$ does not stop the optimization, and 2) an easier subset of images can be chosen by the optimizer. The first problem is a standard mismatch between minimizing a loss and maximizing accuracy. In our case it can mean that the optimizer can over-focus on a particular image that has already been misclassified correctly as

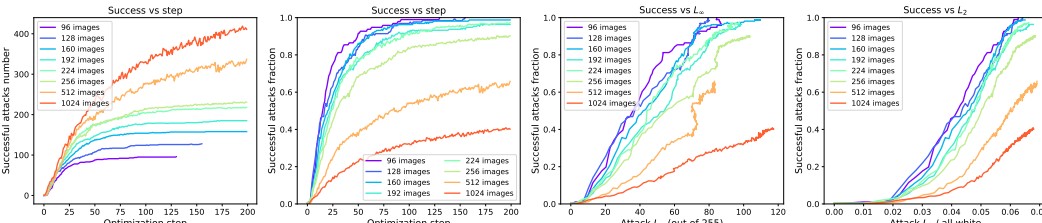

Figure 5: Attacks on different numbers of $224 \times 224$ images at once. Large batches get more successful attacks in the 200 optimization steps we ran the experiment for, likely by focusing on the easier images in the batch. For batches of $\approx 160$ and below, we get $100\%$ success for the simultaneous multi-attack.

the target class, while ignoring the ones that have not been successfully attacked yet. To see the effect of batch size, Figure 5 shows the success of attacks on different batch sizes of $224 \times 224$ images after 200 steps of optimization. The larger batches lead to more images successfully attacked, likely focusing on the easier subset of them. Small batches, in this case 160 and below, get 100% success rate.

### 4.3 MULTI-ATTACKS AGAINST ENSEMBLES

We trained 10 independently initialized `SimpleCNN` models on CIFAR-10 and developed multi-attacks against ensembles of subsets of them (averaging their output logits for a given input) at the resolution $32 \times 32 \times 3$. We used 500 steps of the attack optimization at the learning rate of $10^{-2}$ with the Adam optimizer, and checked the number of images we successfully attacked at t he end of the optimization. Each ensemble size is run 3 times and the resulting average and standard deviation are shown in Figure 3. The larger the ensemble, the fewer images we can attack at the same time, which is consistent with a broad trend of ensembling increasing out-of-distribution (Fort, 2022) and adversarial robustness (Tramèr et al., 2020).

### 4.4 STARTING AT NOISE

We experiment with starting with real images $X_1, X_2, \ldots, X_m$ is any different from starting with random noise samples. In Figure 6 we show that, at least visually, the number of successful attacks as a function of iteration and $L_2$ and $L_\infty$ distances seems very similar between real images and noise samples of the same mean and standard deviation. The experiment was done with $224 \times 224$ images, 1024 images in total, and for 200 iterations of a $10^{-2}$ learning rate with Adam. We can take $\mathcal{O}(100)$ samples of noise, and with a single perturbation $P$ change their classification to equally many arbitrarily chosen classes.

In Section A.1 we show that instead of starting from $n$ independent realizations of noise, we can instead start with a single image and add random realizations of noise as a perturbation to it. For a sufficient amount of noise, these images act as distinct for the purpose of designing a multi-attack against them.

### 4.5 MODELS TRAINED ON RANDOM LABELS

The structure of the space of pixels and the way it is partitioned into classes is what allows multi-attacks and adversarial attacks in general to exist. To see what the effect of training on real data and labels vs training on randomly permuted but fixed labels is, we trained a `ResNet50` on CIFAR-10 first with real labels, and then with labels randomly permuted and fixed. As (Zhang et al., 2016) shows, a large enough network can fit to 100% precision a training set with randomly assigned labels. Such learning is, however, distinct from training on semantically meaningful labels in many ways, and in our experiments we see a clear signal that models trained on random random labels are more susceptible to multi-attacks. Figure 7 shows the result of multi-attacks with 500 optimization steps at the learning rate of $10^{-2}$ against a batch of 128 $32 \times 32$ images of CIFAR-10. Models trained on

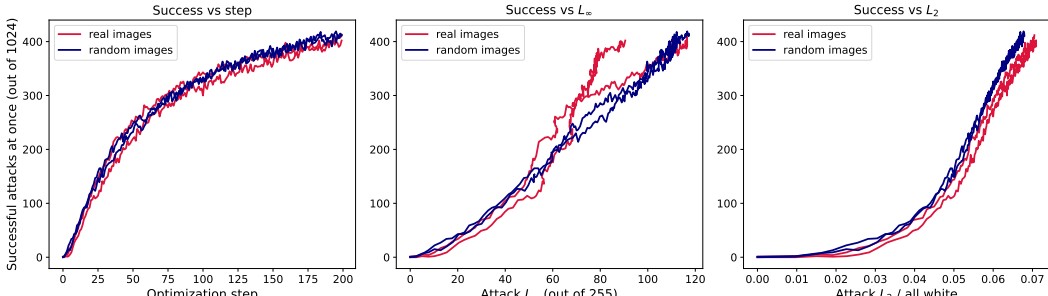

Figure 6: The number of successfully attacked images out of 1024 randomly chosen but fixed samples of CIFAR-10 with 1024 randomly chosen but fixed ImageNet (1000 in total) classes as a function of the size of the adversarial perturbation as compared to images of random noise of the same mean and standard deviation. Real images and noise samples do not differ in their susceptibility to multi-attacks against them. The plots show two random experiments for each type.

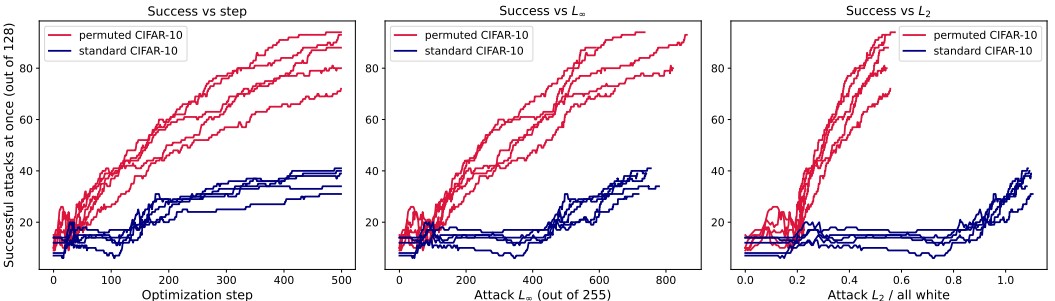

Figure 7: Multi-attacks against ResNet50 trained on real vs random labels of CIFAR-10. Models trained on random labels are easier to attack and allow for multi-attacks on more images at once.

random labels are easier to attacks given a perturbation strength, and also allow for more simultaneous attacks.

### 4.6 LONG LINES OF ADVERSARIES

The very high number of distinct high-confidence regions surrounding each image allows for simultaneous satisfaction of many constrains. A particularly interesting situation is to attack a series of images that all derive from the same original, $X_0$, but to which the adversarial perturbation is progressively applied with an increasing multiplicative factor.

$$(X_0, P, m) : (X_0 + P, X_0 + 2P, X_0 + 3P, \dots, X_0 + mP). \tag{5}$$

In a geometric sense, we are creating a straight line through the pixel space, starting at an image $X_0$ and gradually, in integer steps of $P$, moving in the $\hat{P}$ direction. The $i^{\text{th}}$ image in the line is $X_0 + iP$ and we can optimize $P$ such that it gets mapped to a desired class $c_i^*$ of our choosing, $\arg\max f(X_0 + iP) = c_i^*$. These target classes can be arbitrary, or we can choose them all to be the same target class. If we choose them to be the same, we would have identified a direction $P$ in which the scale of the adversarial attack preserves its function as discussed in Section 4.7

Figure 8 demonstrates an attack, $P$, that starts at an image, $X$, of an airplane and gradually changes the classification to classes 111, 222, 333, 444, and 555 for $X + P$, $X + 2P$, $X + 3P$, $X + 4P$, and $X + 5P$ respectively. The class decision in between the integer values of the $\alpha P$ is filled with the respective classes, showing that the attack is not fragile to small changes of $\alpha$ outside of the directly optimized for integer multiples. The attack shown is to a $32 \times 32$ image, and therefore the magnitude of the perturbation is large. Were we to use $224 \times 224$, the perturbation would be much less prominent, as discussed in Section 4.1. Figure 9 shows a second case of such a line of adversaries, this time at a higher resolution and lasting for 9 steps.

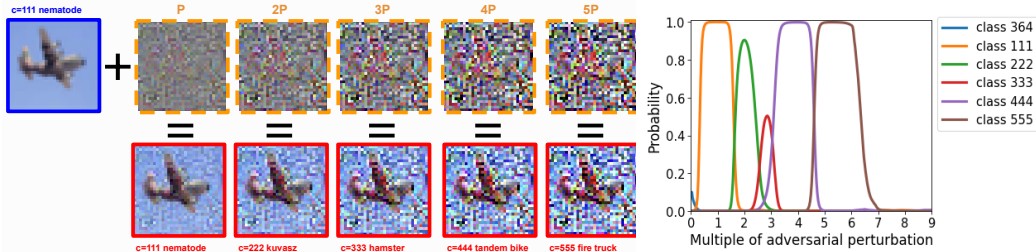

Figure 8: The same attack, $P$, when applied with different strength, leads to the image $X$ being classified as classes 111, 222, 333, 444, and 555 for $X + P$, $2P$, $3P$, $4P$ and $5P$ respectively. The figure on the left shows the resulting attack perturbations and the corresponding perturbed images, while the right-hand side shows the probabilities of the classes (including class 364, the original class of $X$) as a function of the strength of the perturbation. The x-axis shows $\alpha$ for $f(X + \alpha P)$

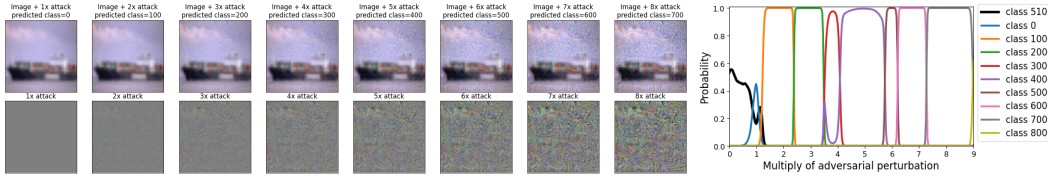

Figure 9: The starting image $X_0$ is classified as class 510 (*container ship*). We optimized an attack $P$ that makes $X_0 + P$, $X_0 + 2P$,…,$X_0 + 10P$ classified as classes 0, 100, 200, 300, ..., 900. On the left panel, examples of progressively more corrupted images are shown that are classified as such. On the right panel, the probability of the target classes are shown, peaking around the multiples of the adversarial attack that corresponds to them.

## 4.7 SCALE-INDEPENDENT ATTACKS

A particular consequence of being able to find lines of adversarial attacks is that we can find such lines where the target class does not change with the scale $\alpha$ of the attack $\alpha P$. Figure 10 shows the result of such an attack. A picture of a bird is attacked with a perturbation $P$ that is optimized such that $X + P$, $X + 2P$, …, $X + 60P$ are all classified as class 111 (nematode). The right panel of Figure 10 shows that the class decision holds in between the integer values of $\alpha$, and also that, while only directly optimized up to $X + 60P$, the predicted class stays at 111 all the way to $X + 160P$, showing an amount of generalization of this scale-independent attack.

## 4.8 FINDING SHAPES IN THE PIXEL SPACE

Using the very large number of regions surrounding each image in the pixel space, we decided to optimize for pairs of attacks $P_x$ and $P_y$ that, together with a starting image $X_0$, define a two-dimensional affine space (a "plane") in the pixel space. This is related to the cutting-plane method used in (Fort et al., 2022), however, there the bases are randomly chosen (and as such $\mathcal{O}(10)$ are needed to find high-confidence adversarial attacks). Here, we optimize for $P_x$ and $P_y$ such that the affine subspace they trace out in the space of pixels spells out words and draws images of our choosing, demonstrating the flexibility of the input-space partitioning of deep neural networks.

(Skorokhodov & Burtsev, 2019) shows that in the space of weights and biases of deep neural networks (also known as the loss landscape), there exists a vast richness of two-dimensional sections where the loss value traces various shapes and images, and where we can optimize for finding particular ones, such as the shape of a *bat*, a *skull*, a *cow*, or the planet *Saturn*. (Czarnecki et al., 2020) extends this work and show that this a general property of sufficiently large and deep neural networks.

Inspired by this experiment and given the rich nature of the class boundaries in the space of pixels that we demonstrated by creating *multi-attacks* on many images and towards many labels at once, we find similarly suggestive shapes in the space of images and their classification decisions.

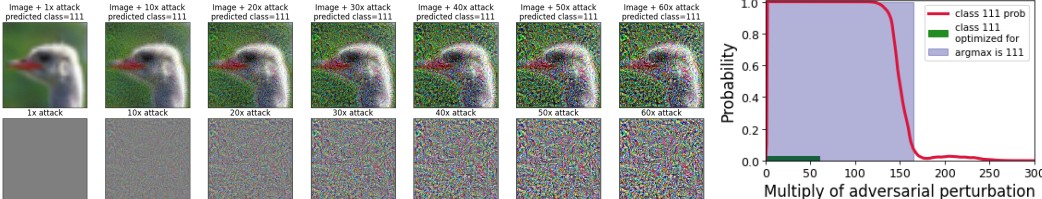

Figure 10: The starting image $X_0$ is classified as class 98 (*red-breasted merganser*). We trained an attack $P$ that makes the $X_0 + P$, $X_0 + 2P$,...,$X_0 + 60P$ classified as class 111 (*nematode*). On the left panel, examples of progressively more corrupted images are shown. On the right panel, the probability of the target class 111 is shown. The class 111 remains the highest for the full 60 multiples of the attack, and continue to be so until ≈160 multiples, demonstrating an amount of generalization.

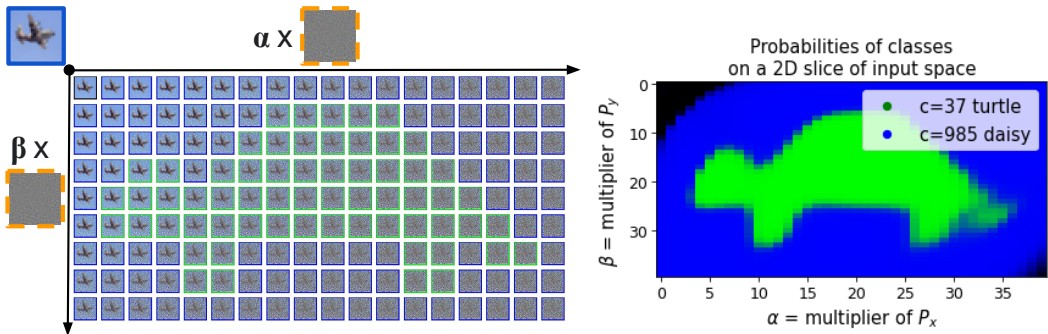

Figure 11: A picture of a tortoise drawn by the ImageNet class c=37 *turtle* with a background of class c=985 *daisy*, found on a two-dimensional slice of the space of inputs around a CIFAR-10 image of an airplane. By finding two adversarial perturbations $P_x$ and $P_y$, we were able to optimize for these classes on a two-dimensional slice of the pixel space starting at the plane image $X_0$. This further demonstrates the richness and flexibility of the class partitioning of the pixel space.

Starting at an image $X_0$, we optimize for two adversarial attacks, $P_x$ and $P_y$, defining a two-dimensional affine subspace $X_0 + \alpha P_x + \beta P_y$. For integer values of the parameters $\alpha = 0, 1, \ldots, W - 1$ and $\beta = 0, 1, \ldots, H - 1$, we force the classifier to classify $X_0 + \alpha P_x + \beta P_y$ as the class specified by a target image $T[\alpha, \beta]$. $T[\alpha, \beta]$ is a bitmap of the image we would like to discover in the pixel space in the neighborhood starting at the image $X_0$, specifying a target class for each $(\alpha, \beta)$. In Figure 2 we show a random illustration of a particular section of the space of inputs. Here, we optimize its orientation to look in a particular way.

In Figure 1b we show a result of this optimization: the word *AGI* spelled in the ImageNet class *laptop*, an outline of a tortoise drawn in the class *turtle* and the background of this image being made of the class *daisy*. Figure 11 shows the grid of images $X_0 + \alpha P_x + \beta P_y$ spanned by the two trained adversarial attacks $P_x$ and $P_y$.

## 5 DISCUSSION AND CONCLUSION

In this work, we have demonstrated the existence of *multi-attacks* – we show that we can take a large number of images (e.g. $> 100$ for images of the $224 \times 224$ resolution and an ImageNet-trained classifier) and optimize for a single adversarial perturbation $P$ that we call a *multi-attack* and that, when added to each of the images, makes them misclassified as arbitrarily chosen target classes. Given this, we estimate the number of distinct high-confidence class regions in the pixel space around every image to be approximately $10^{\mathcal{O}(100)}$ using a simple toy theory, which links the maximum number of simultaneously attackable images $n_{\max}$ and the number of classes $C$ to the number of regions $N$ as $N \approx \exp\left(n_{\max} \log(C)\right)$.

Exploiting this flexibility, we show that we can find two-dimensional sections of the pixel space that trace words and images in whatever class we choose. This can be understood as a consequence of the sheer number of two-dimensional patterns that hide in a high-dimensional space partitioned into $10^{\mathcal{O}(100)}$ cells reachable from each point. We can also find adversarial attacks $\alpha P$ that change the image attacked to a different class based on their strength $\alpha$ and that can change them to the same class regardless of the scale $\alpha$ for a range of $\alpha$s.

We show that classifiers trained on randomly assigned labels are easier to design multi-attacks against as compared to classifiers trained on labels that are semantically connected to their corresponding images. Interestingly, whether we are modifying real images or samples of noise does not seem to have any effect on how easy it is to design a multi-attack.

Every strategy designed to defend against adversarial attacks has to deal with this issue: around each image there are very many, e.g. $10^{100}$ neighboring images that, to a human, differ only slightly from the original image, and yet are classified very different by a learned classifier. Making sure that each of these gets classified correctly is a difficult task. For example, it is virtually impossible to add all of them to the training set with the correct (to a human) label, as some strategies attempt to do. Unless we can somehow do this exponentially effectively, defending against attacks might be a very hard task. The key problem is the dimensionality of the input space and how small a cell each high-confidence region seems to be.

Our findings open several directions for future work. More rigorous theory is needed to tightly characterize the scale of this redundancy. Study of failure cases and images resistant to multi-attacks could inspire new defensive techniques. And development of fast algorithms to find minimal multi-attacks could enable applications.

By exposing the scale of redundancy in neural network classifiers, we hope these results will inspire further investigation – both of the root causes of this extreme flexibility, and of potential solutions to make models more robust.

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

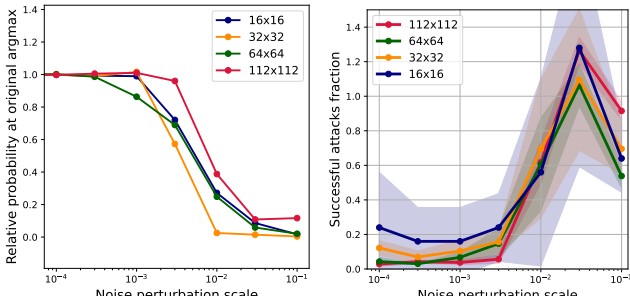

Figure 12: **Left panel:** The relative probability of the original argmax class of images perturbed by Gaussian noise of different standard deviations. **Right panel**: The number of successful attacks on a batch of 64 images comprising the same image + noise of different scale (x-axis) towards 64 randomly selected target classes, divided by the number successful attacks starting from 64 distinct images. If the perturbation is too small, we basically have 64 copies of the same image and cannot get it to change towards 64 random classes with the same perturbation. If we go far enough ($\sigma \approx 3 \times 10^{-1}$), the noisy images work as well as 64 random images.

# A  APPENDIX

## A.1  INDEPENDENT IMAGES VS NOISE PERTURBATIONS OF A SINGLE IMAGE

We wanted to understand at what scale of a noise perturbation $n$ do noisy versions of the same image $X_0$ act as independent in the sense that we can optimize each realization of the noise added to the image towards a different class. Instead of independent starting images $X_1, X_2, \ldots X_n$, we take the same image and add different realizations of noise drawn from a Gaussian distribution with 0 mean and standard deviation $\sigma$ as $X_i = X_0 + s_i$ for $s_i \sim \mathcal{N}(0, \sigma)$.

For a small perturbation $\sigma$, the resulting images are essentially the same and we therefore cannot change them 64 different classes with a single perturbation. However, if $\sigma$ is sufficiently large, the 64 images effectively behave as if they were 64 random, distinct images and we are be able to convert each of them to a different class with a multi-attack. In Figure 12 we show the fraction of the images we were able to attack successfully after 20 optimization steps starting from $X$ with different realizations of the noise, divided by the number we reach starting from 64 random images. For $\sigma \approx 3 \times 10^{-1}$ we reach parity. In other words, taking a single image $X$ and adding a normal noise from $\mathcal{N}(0, 3 \times 10^{-1})$ effectively renders the perturbed images independent for the purposes of simultaneous adversarial multi-attacks. Unfortunately, on the left panel of Figure 12 we also show that by that point, the images are so noisy that the probability in the original argmax class drops to effectively zero. These effects seem to be independent of the image resolution. (Fort et al., 2021) generates multiple differently augmented copies of the same image within a batch and finds that this increases the speed of training as well as the final accuracy on several image classification benchmarks, which is a related idea.