# OpenReview forum: "Multi-attacks: A single adversarial perturbation for multiple images and target labels"
_ICLR.cc/2025/Conference — Submitted to ICLR 2025_

### Official Review · Reviewer_qQ9n · 2024-10-21

**Soundness:** 2
**Presentation:** 2
**Contribution:** 2
**Rating:** 3
**Confidence:** 4

**Summary:**

The authors study a special kind of adversarial example designed to be effective for multiple images and multiple targets. The authors conducted a series of experiments to understand the properties of the attack, such as its limit, effect on different models, etc.

**Strengths:**

(1) The authors conducted a wide range of experiments relating to the attack.

(2) Some interesting phenomena are observed. For example, the number of successfully attacked images scales linearly with the logarithm of the resolution, as $n_{max}$ ∝ log r.

**Weaknesses:**

Major:

(1) The paper is organized poorly. The logical connection between sections is weak and it feels like reading a daily research log rather than a paper ready to be published. It is highly not recommended to fill a paper with 6 pages of toy experiments without clear connections between them and the necessity to conduct each of the experiments.

(2) There is no novel technique and no novel attack scenario ( with the universal adversarial attack well known by the community) proposed. Besides, I don't think the observed phenomena are interesting to me since it is trivial that the attack will be stronger when you use more optimization steps, perturbation budge, and larger batch size. Scaling law is interesting and appealing, but not every scaling view is meaningful.



Minor:

(1) The experiment setting is incomprehensive. The authors only considered 2 ResNet models and only considered the ImageNet dataset. I would like to see some of the claims verified in a broader sense.

(2) The authors should pay more attention to explaining the significance and importance of some of the experiments.

(3) The text displayed in some figures is way too small. And some figures are not clear and informative enough.

**Questions:**

Please see the weakness part.

---

### Official Review · Reviewer_x2r2 · 2024-10-22

**Soundness:** 1
**Presentation:** 3
**Contribution:** 2
**Rating:** 3
**Confidence:** 4

**Summary:**

This paper proposes the "Multi-attack": a novel formulation of adversarial attack. In the Multi-attack, the attacker is given a classifier $f$ to attack, a set of samples (in the experiments, images) $X_1,...,X_n$ to attack, and a set of target classes $c^*_1,...,c^*_n$. The goal is for the attacker to produce a single perturbation $P$, such that for _all_ $i$ from 1 to n, $X_i + P$ is classified by $f$ as $c^*_i$. In other words, the attacker must attack all samples simultaneously, to achieve different targets.

The paper also provides (In Section 3) a high-level sketch of a theoretical argument, claiming that the existence of a successful multi-attack perturbation  $P$ for $n$ simultaneous samples within some bounded area (e.g., within a norm-constrained ball) implies with high probability that on average, there must exist $\gtrsim C^n$ "cells" -- that is, distinct continuous regions where the classification of f(x) remains constant --  within the perturbation-budget area around each sample $X_1,...,X_n$ (where $C$ is the number of classes). Because multi-attacks on CIFAR-10 classifiers are claimed to be demonstrated with $n = \mathcal{O}(100)$, it is claimed that this implies that there are $10^{ \mathcal{O}(100)}$ such "cells" around each CIFAR-10 image, a finding which is claimed to demonstrate the richness and complexity (but also adversarial vulnerability) of neural network decision boundaries.

Experiments include:
- Running Multi-attacks on Imagenet classifiers with upscaled CIFAR-10 images, and observing the success rate  (fraction of samples out of $n$ for which $X_i + P$ is classified as $c_i^*$) and L_2 and $L_\infty$ norms of these attacks for varying numbers of attacked samples $n$ and amounts of upscaling (i.e, resolution at which the attack was generated) -- Fewer samples are easier to attack, and higher resolution is easier to attack
- Measuring the affect of ensembling on multi-attack success rates for CIFAR-10 ensemble classifiers with CIFAR-10 samples.  -- Larger ensembles are harder to attack
- Comparing multi-attack performance on Imagenet classifiers of random noise samples to the performance on upscaled CIFAR-10 images. -- Approximately no difference
- Comparing  multi-attack performance on CIFAR-10 samples with CIFAR-10 classifier versus CIFAR-10 classifiers with random labels. -- Classifier trained on random labels is easier to attack

The paper also explores other kinds of highly-constrained attacks, such as:
- "Line attacks" where, starting at sample $X$, the samples $X + aP$ for different scalar values of $a$ have different targetted classifications.
- "Line attacks" where, starting at sample $X$, the samples $X + aP$ for different scalar values of $a$ have the same classification -- it is shown that the classification continues along the line for values of $a$ beyond the largest optimized value of $a$, and also that the classification persists between the values of $a$ that were directly optimized
- "Grid attacks" where, starting at sample $X$, the classifications of the samples $X + aP_1 + bP_2$  for scalars $a$, $b$ form a targeted two-dimensional "image" in a grid.
These additional experiments are run on only one or a handful of samples each.

**Strengths:**

The multi-attack framework is, to my knowledge, novel, and, if successfully demonstrated, would underscore the extreme richness of neural network decision boundaries in high dimensions.

The writing is clear and easy to follow.

The additional demonstrations, such as the grid attack, are interesting to ponder.

**Weaknesses:**

While the paper introduces some interesting ideas, it is difficult to identify a clear goal that the paper successfully accomplishes. The multi-attack is not a practical adversarial threat model: unlike the similar Universal Adversarial Perturbations (Moosavi-Dezfooli et al. 2017), in which a single attack perturbation is optimized to target the same class for any image, there is no generalization to novel images in the multi-attack. Access to the classifier, significant compute, and "edit-access" to all attacked samples is still required, as in standard adversarial attacks. Therefore, it is hard to imagine why an attacker would need to attack in this way.

In any case, the multi-attack is not really presented as a practical attack; instead, it is claimed that the existence of successful multi-attacks tells us something novel and interesting about deep networks. The paper fails at this in two ways:

- The theoretical argument, that the multi-attack implies the existence of many distinct "cells" with different classifications in the neighborhood of each sample, is flawed.

- It is not convincingly demonstrated that imperceptible, or even near-imperceptible, multi-attacks are possible in a natural setting, where the attacked samples are drawn from the same distribution that the classifier was trained on.

# Theoretical claims (Section 3)

I am not convinced by the argument in this section, and think that it is flawed. In this section, the authors consider $n$ samples $X_1,...,X_n$, each with a target class $c^*_1,...,c^*_n$, and an attacker that is trying to perturb all samples into the target classes at once, using the same perturbation vector $v$ on each sample. Each sample $X_i$ is assumed to be surrounded by N "cells"-- that is, continuous regions where the classification of f(x) remains constant -- within the perturbation budget.

The basic argument made is that a given random perturbation $v$ has a $\sim (1/C)^n$ chance of achieving the desired target classification for all $X_i$'s simulaneously (under some implied i.i.d. assumption). Because the classification $f(X_i +v)$ is constant within each "cell", there are effectively only $N$ "chances" for this to work, so the overall probability of such a perturbation vector $v$ existing at all is $\sim N(1/C)^n$. Then, if multi-attacks are consistently successful, it implies that there must exist $N \gtrsim C^n$ "cells" around each sample. (This is similar to a "uniform convergence" argument in classical machine learning, where each "cell" is a hypothesis.)

However, this argument assumes that the "cells" around each sample in $X_1,X_2,...,X_n$ _line up with each other exactly_. In reality, this is very unlikely to hold. For example, there may be two perturbation vectors $v$ and $v'$, such that $X_i +v$ and $X_i + v'$ belong to the same "cell" around $X_i$, but  $X_j +v$ and $X_j + v'$ belong to different "cells" around $X_j$. Then the number of perturbations $v$ that yeild distinct classification patterns can potentially be much larger than the number of regions of constant classification around each individual sample $X_i$.

To demonstrate this, and show how the argument can fail, consider the following setting, for binary classification ($C = 2$):
- The classifier $f$ is a decision tree, where each node tests the value of a single dimension of $X$. Therefore, the local decision boundaries are axis-aligned planes.
- The attack budget/"nearby region" is an $L_\infty$ ball of radius $r$  (inclusive of the surface of the ball) around each sample
- The classifier is "locally linear" within each ball around each $X_i$. In other words, only one planar decision boundary appears within the $L_\infty$ ball around each $X_i$.   This means that the true value of N is $N=2$.  Note that this also implies that each sample individually is vulnerable to an $L_\infty$ adversarial attack of magnitude $r$. Furthermore, we assume that the directions of these local linear classifiers are uniform and i.i.d. for each $X_i$.

Then, in this setting, a multi-attack exists within the attack budget for all n samples simultaneously as long as no two samples $X_i, X_j$ are near decision boundary planes in the same dimension, but where moving $X_i$ to $c_i^*$ requires perturbing $X_i$ in the opposite direction as perturbing $ X_j$ to $c_j^*$. We can upper-bound this failure rate by simply the probability some $X_i, X_j$ having decision boundaries of the same dimension: this is the standard "birthday problem", and so no such pair is likely to exist as long as $n \lesssim \mathcal{O}(\sqrt{d})$. Then, a multi-attack *can* exist for  $n = \mathcal{O}(\sqrt{d})$, so the theory discussed in the paper would suggest that there exists $\sim 2^{\sqrt{d}}$ "cells" around each sample. However, there in fact are only two "cells" around each sample, because the classifier is locally linear.

Additionally, in the explanation of the argument, the "cells" are defined as "a high-confidence class area for some class."  (Line 135) If the overall argument can be fixed, it would be better to say "an area with maximum logit c, for some $c \in [C]$". Whether or not the confidence of a prediction is high is typically not considered when determining whether an attack is successful (e.g., in the experiment section of this paper), so "high-confidence" should not be relevant. Overall, it would be better to make the argument more rigorous (again, if it can be fixed or clarified so that it is correct), by making the simplifying assumptions more explicit.



# Experimental Claims

In the "top-line" experiments (Figures 4 and 5; as well as Figures 8, 9, 10, and 11), CIFAR-10 images (and random Gaussian images) are used with an ImageNet classifier. This does not represent the usual setting for studying adversarial attacks (where the samples are generally assumed to come from the same distribution that the classifier was trained on), and may significantly afffect the results. It is possible, for example, that adversarial vulnerability is greater or otherwise different for images far from the training data manifold (i.e., naively upscaled CIFAR-10 images for an ImageNet classifier) than it is in the usual setting where the training and test data are from the same distribution, so this is a significant experimental confounder.

In fact, in some of the experiments where the training and testing set **are** actually the same (Figure 7, where the purple lines represent a  network that is trained and tested on CIFAR-10 at standard resolution) we see that the images are **not** significantly vulnerable  to "multi-attack." In fact, for $L_\infty$ perturbations as large as 255/255, we still see that < 20/ 128 images are successfully attacked for all seeds. Note that this is a 10-class problem, so  results almost this good could be obtained at this huge perturbation size by simply targetting the plurality class of $c_1^*,...,c_n^*$. (If clipping was applied after adding the perturbation in order to make the images occupy a valid range, the attacker at this perturbation size could set each pixel of all samples to an arbitrary {0,1} value, making targetting a single class trivial. I don't think such clipping was applied in this paper, but that's a problem in itself; see below.) The other "natural" CIFAR-10/CIFAR-10 experiment is for ensemble classifiers (Figure 3), but here, the results for the attacks against ensembles do not indicate the norms of the attacks at all.


In all of the experiments, the attack optimization is totally unconstrained, with no regularization to stay near the zero perturbation. While this produces valid results for some experiments, where the success rate is plotted against the observed perturbation norm at different steps of the optimization, it is likely not the most effective attack for producing minimal-norm adversarial perturbations. It would be better to use something like a Carlini & Wagner attack to find the maximum success rate for a given norm-bound.


In several results (Figures 4-2 and 7-2) The $L_\infty$ attack radii go far above 255, suggesting that there are no sort of box constraints on the attack to make the images valid inputs to a classifier. While I understand that this could be tricky (because the box constraints would be different for different images), I think this should be handled in some way to make the attack meaningful in a practical threat model. For example, one could clip $X_i + P$ to the box constraints once the attack is added. (An alternate approach would be to develop multi-attacks that are "patch" adversarial attacks [That is, the attacker fully determines a bounded patch of pixels on each image.] This would eliminate the issue of incompatible box constraints. In fact,  prior work has been done on "universal" patch attacks, which are similar to multi-attacks, but with a single target class. See  Moosavi-Dezfooli et al. 2017)

It is unclear what the units "Attack L2 / all white" mean in the figures. Does this mean that the L_2 magnitude of the attack is normalized by the magnitude of an "all white" ([255,255,....]) image? If so, the attack perturbations are extremely large compared to what is typically encountered in adversarial robustness literature. I would like to see sample images with perturbation magnitudes labeled to provide a sense of how "imperceptible" these perturbations are. (No sample image currently in the paper is labeled with its absolute attack magnitude, so these could be outliers with unusually small-magnitude perturbations)

Line 201-204: "Interestingly, it seems (by visual inspection alone) the number of successfully attacked images scales linearly with the logarithm of the resolution, as n max ∝log r(as shown in the right-most panel of Figure 4.":  I don't find this observation very meaningful, because the attack optimization is unconstrained, so success or failure  might be just depend on the nature of the optimizer, rather than the actual decision boundary. (Also, Missing close-paren.)

Sections 4.6-4.8: These sections show results for only one or two individual images per experiment, and therefore give no indication as to how well these techniques work in general. (Also, again, scaled-up CIFAR-10 images are being used with an Imagenet classifier.)

In fact, for the other experiments in the paper, only a small number (1 to 5) "multi-attack" optimizations are performed for each parameter setting. While each attack involves many samples, it would be better to show a larger number of independent runs of the algorithm.


Carlini, N., & Wagner, D. (2017). Towards evaluating the robustness of neural networks. In 2017 ieee symposium on security and privacy

Moosavi-Dezfooli, S. M., Fawzi, A., Fawzi, O., & Frossard, P. (2017). Universal adversarial perturbations. In Proceedings of the IEEE conference on computer vision and pattern recognition (pp. 1765-1773).

# Minor Issues

- Line 40: "simultaneous attack" -> "simultaneously attack"

- Use of capitals for specific images X and perturbations P is a bit odd/nonstanard, becasuse these are not random variables

- Eq. 1: Notation is inconsistent with what is introduced above. $y^*_i$ should be $c^*_i$, and it is inconsistent whether i ranges from 1 to n or 0 to n-1

- Line 148-149: "the difference in the order of magnitude is small": More accurate to say: "the difference in the order of magnitude _of the exponent_ is small": we're talking about the difference between 10^100 and 10^300.

- Line 157: "we would first change" -> "we first changed"

- Line 241: "t he" -> the

- Line 249-250: "We experiment with starting with real images X1,X2,...,Xm is any different from starting with
random noise samples.": check grammar

- Line 307: "constrains" -> constraints

- Figure numbering: Figure 4 comes before, and is reference before, Figure 3

- Line 446-447: ". For example, it is virtually impossible to add all of them to the training set with the correct (to a human) label, as some strategies attempt to do. ": What strategies?

- References: Only 4 of the references have a venue listed. While some of the others are pre-prints, some are not (e.g., Zhang et al, ICLR 2017); even for pre-prints, something, e.g., Arxiv  should be listed as a venue.

**Questions:**

- Is there a practical threat model corresponding to the multi-attack?

- Are there any general take-aways we should get from these experiments, other than the "cell" argument in Section 3?

- How do multi-attacks fare on ImageNet samples on an ImageNet classifier? (This shouldn't be more computationally expensive than attacking upscaled CIFAR-10 images.)

- How do multi-atttacks fare when the attacked samples are constrained to be valid images? (As discussed above, this involves imposing box constraints on $X_i+P$, which could involve either clipping or switching to a patch-attack framework; also, attacks should ideally obey image quantization constraints; see Carlini & Wagner 2017, section  5-D)

- For a given norm constraint, does a norm-regularized optimization (such as a Carlini-Wagner attack) result in a more successful multi-attack?

- What are the norms of the shown attack images? Can images be added for the typical magnitudes of the attacks shown in the paper?

- What does  "Attack L2 / all white" mean in the figures?

- What are the norms of the attacks in Figure 3?

- It would be better to show results for averages of many independent runs of the multi-attack, rather than just a few runs. Also, it would be good to see statistics on multiple runs of the other constrained attacks (Sections 4.6-4.8) on many randomly-chosen sample images.

---

### Official Review · Reviewer_H9F2 · 2024-10-29

**Soundness:** 1
**Presentation:** 2
**Contribution:** 1
**Rating:** 1
**Confidence:** 5

**Summary:**

This paper introduces multi-attacks, where one perturbation changes multiple images into different target classes. Higher-resolution images and models with random labels are more vulnerable, while ensembles offer better defense. The study highlights the challenge of defending against these attacks due to the vast number of high-confidence regions in pixel space, urging future research on more robust defenses.

**Strengths:**

* Introduces multi-attacks to manipulate multiple images with a single perturbation.
* Evaluates attacks on datasets like CIFAR-10 and ImageNet using architectures such as ResNet50 and ResNet18.
* The theory part is nice.

**Weaknesses:**

* The paper lacks a dedicated related work section.
* The proposed method lies between two adversarial attack approaches:
  - Standard adversarial attacks: where each image is attacked individually.
  - Patch attacks: where a malicious patch is added to the image.
  The authors should include comparisons with methods from both of these fields.
* The y-axis of the graphs is inconsistent and should use accuracy (acc) throughout.
* The authors should evaluate the approach using adversarially trained classifiers.
* The perturbation should be bounded as commonly accepted in the literature.
* It is not surprising that batch size affects optimization.
* The paper lacks a main results section with comparisons to other methods and focuses primarily on ablation studies.
* The allowed perturbation exceeding 255 under \(L_\infty\) norm is questionable and should be addressed.

**Questions:**

See weaknesses

---

### Official Review · Reviewer_XKrc · 2024-11-04

**Soundness:** 3
**Presentation:** 3
**Contribution:** 2
**Rating:** 6
**Confidence:** 4

**Summary:**

The paper introduces a novel adversarial attack method termed "multi-attacks," which involves generating a single perturbation that can target multiple images simultaneously. The authors explain that this approach is feasible due to the vast number of adversarial regions surrounding each image. This abundance allows for the incorporation of several additional constraints into the adversarial example search problem, making it relatively straightforward to identify a perturbation that satisfies all criteria. They also provide a simple theoretical framework to elucidate this phenomenon and discuss various factors that affect the ability to discover such multi-attack perturbations.

**Strengths:**

Novelty: The paper introduces a compelling and innovative approach by applying extensive constraints to the adversarial problem. This perspective challenges the common belief that adversarial examples are rare and unique, demonstrating instead that a multitude of such examples exists across various regions.

Ablation Studies: The authors conduct several ablation studies, varying key factors such as the model architecture or ensembles, the number of images attacked simultaneously, and the resolution. This thorough exploration enhances the validity of their findings and provides deeper insights into the dynamics of multi-attacks.

**Weaknesses:**

Insufficient Theoretical Framework: The paper lacks a robust theoretical foundation, as many concepts presented as theories could be empirically tested. For instance, the impact of a random perturbation (mentioned in line 134) raises questions about whether there are numerous distinct regions or simply one large subspace. Additionally, it would be insightful to explore how modifying the multi-attack loss to avoid focusing on easier subsets of images (as discussed from line 231 onward) might alter the results. A more rigorous theoretical underpinning, even if empirically based, would significantly enhance the paper.

Choice of Network and Dataset: The authors utilize the CIFAR-10 dataset with an ImageNet classifier without providing a clear rationale for this choice. Furthermore, the explanation of how adding noisy pixels affects the "resolution" of the images is not adequately addressed.

Unsubstantiated Claims in Conclusion: The conclusion mentions applications to robust training without providing sufficient justification or experimental support for these claims. This lack of evidence weakens the overall impact of the paper’s findings.

Figure ordering: hard to follow.

**Questions:**

1. Why are there multiple adversarial regions instead of just a single large one? What factors contribute to the existence of these distinct regions?

2. What is the absolute size of the perturbations identified in the L2 norm? How do these sizes compare across different examples?

3. How does the multi-attack perturbation differ from simply using the sum of all individual adversarial perturbations found for each image? What advantages does the multi-attack approach offer?

---

### Meta-Review · Area_Chair_RVPb · 2024-12-17

**Metareview:**

Summary: This paper studies the problem of designing a single adversarial perturbation to attack multiple images. The first principle behind the design is due to the authors' belief that there are vast number of adversarial regions surrounding each image, which make the incorporation of several additional constraints into the adversarial example search problem possible.

Strength:
1. The problem of designing a single adversarial perturbation to attack multiple images is important in the adversarial community. This can tract back to patch attacks and universal attack.
2. Ablation study such as the model architecture or ensembles is provided.
3. The authors try to analyze the problem by theory.

Weakness:
1. Reviewers are not convinced by the argument made in this paper.
2. Reviewers have concerns on the novelty of the paper.
3. Reviewers believe the paper is not of well organized.
4. Experiments are only with CIFAR-10 and ImageNet datasets and classifiers. Experiments on more datasets and settings are expected from the reviewers.

As most reviewers vote against the paper with (even strong) rejection and the authors do not provide a rebuttal, AC believes the paper is of a clear rejection.

**Additional Comments On Reviewer Discussion:**

No rebuttal is provided by the authors, so there is no discussion between authors and reviewers.

---

### Decision · Program_Chairs · 2025-01-22

Reject